# Reproductive Systems, Transfer and Digestion of Spermatophores in Two Asian Luciolinae Fireflies (Coleoptera: Lampyridae)

**DOI:** 10.3390/insects12040365

**Published:** 2021-04-20

**Authors:** Xinhua Fu, Lesley Ballantyne

**Affiliations:** 1College of Plant Science and Technology, Huazhong Agricultural University, Wuhan 430070, China; 2Hubei Insect Resources Utilization and Sustainable Pest Management Key Laboratory, Wuhan 430070, China; 3Firefly Conservation Research Centre, 1 Shizishan Street, Wuhan 430070, China; 4School of Agricultural and Wine Sciences, Charles Sturt University, P.O. Box 588, Wagga Wagga 2678, Australia; lballantyne@csu.edu.au

**Keywords:** nuptial gifts, spermatophore, sexual dimorphism

## Abstract

**Simple Summary:**

During mating, fireflies may transfer sperm to the female wrapped in food materials (spermatophores). We investigate in two firefly species structures in both male and female that indicate production and receipt of spermatophores. Their structure, how they attach and discharge the sperm inside the female, and how long they persist might indicate if these fireflies mate only once or more. Potential differences between males and females of both fully winged and species with flightless females are determined. An overview of present knowledge of female reproductive anatomy is given. An argument is mounted suggesting direct observation of biological structures like spermatophores might be the only reliable way of determining their presence.

**Abstract:**

The internal reproductive anatomy of males and females of two Asian Luciolinae fireflies *Emeia pseudosauteri* (Geisthardt, 2004) and *Abscondita chinensis* (L., 1767) is described, and the time course for spermatophore transfer and digestion examined. *E. pseudosauteri* is sexually dimorphic, with a flightless female, and *Abs. chinensis* is sexually monomorphic, with the female flighted. Both are monandrous. Possible female accessory glands are described for the first time for both species. An overview of present knowledge of female reproductive anatomy in the Luciolinae reveals males of 18 species in 10 genera may produce spermatophores and permits speculation about spermatophore production in another 16 genera.

## 1. Introduction

The process of copulation in insects often involves the exchange of a variety of materials from male to female. These may be exchanged prior to, during, or after the process of copulation and the many different forms include food items and spermatophores. The spermatophore is a container that serves to protect the semen during transfer to the female [1]. It is produced by male accessory glands in insects as a viscous substance that surrounds the spermatozooids and solidifies on them. During mating in Luciolinae fireflies, it is extruded as a gelatinous mass through the ejaculatory orifice of the median lobe of the male aedeagus into the female reproductive tract. Often referred to as ‘nuptial gifts’, spermatophores can provide nutrients, water and often defensive chemicals that can improve recipient fitness [2,3,4,5]. There is also the alternative possibility that spermatophores transferred to the reproductive tract may be more likely to manipulate recipient physiology and reduce recipient fitness, driven by sexual conflict [6,7].

Fu et al. [5] found that females of *Aquatica ficta* are monandrous. Given that they also demonstrated that the transfer of a spermatophore to the female provided a longevity benefit for the female, then it would be expected that selection would favour polyandry. The monandrous condition was unexpected. They investigated the time of transfer to the female and subsequent digestion of the spermatophore. Since spermatophore disintegration was complete by 24 h post mating they considered that persistence of an intact spermatophore in the female reproductive tract could not be an explanation for monandry.

Hayashi and Suzuki [8] had indicated that the presence of prespermatophores in the male reproductive tract was a reliable indicator of production of spermatophores and listed eight Luciolinae (including *Pristolycus*) in which males had prespermatophores. South et al. [9] demonstrated spermatophore production in *Luciola* (now *Aquatica*) *lateralis* and *Luciola cruciata*, and Fu et al. [5] in *Aquatica ficta*. South et al. [10] repeated the Hayashi and Suzuki [8] list of eight Luciolinae (including *Pristolycus*) where all males had three accessory glands and were thus considered able to produce spermatophores. Unfortunately, their scoring system led to a possible oversimplification and loss of information as these species with flightless females (*Luciola L. kuroiwae* Matsumura, 1918; *L. parvula* Kiesenwetter, 1874; *L. yayeyamana/filiformis* Matsumura 1918) were all scored as having flighted females. South et al. [10] also indicated a correlation between loss of flight in female Lampyridae and the production of spermatophores. Other studies had postulated that males of species having flightless females do not produce spermatophores [5,11].

We know of only three Luciolinae genera that have flightless females (*Atyphella* Olliff, *Emeia* Fu et al. and some *Luciola* s. str.; [12]). Of these we have no information about *Atyphella* female reproductive anatomy despite there being 11 species having females with reduced flight wings. We considered it necessary to expand the treatment of the Luciolinae especially with regard to species with flightless females. Here we describe the male and female reproductive systems of a dimorphic monandrous firefly *Emeia pseudosauteri* [13], compared with the monomorphic and monandrous *Abscondita chinensis* (L., 1767). Additionally, we examine the process of spermatophore placement and subsequent digestion in the female tract in both species, and reexamine if its persistence, or otherwise, could be an explanation for monandry. An overview of females known from the Luciolinae is given with an indication either of the production of spermatophores, or the likelihood of their production. An argument concerning the difference between scientific accountability and inference with respect to flightless females and spermatophore receipt is mounted.

## 2. Materials and Methods

### 2.1. Study Organisms

*Emeia pseudosauteri* is endemic to China, and has terrestrial trilobite-like larvae predating on small land snails. Adults are sexually dimorphic (Figure 1A–D), and females, which are slightly shorter than the males, have very short hind wings (Figure 1C). Flightless females were observed on grass leaves and stems producing quick single pulse flashes with averaged duration 200 ms and intervals of 340 ms. Flashing males usually flew not higher than 1 m above grass with single short flashes of 590 ms average duration, and intervals of 530 ms [14].

*Abscondita chinensis* is found across China and Taiwan. Larvae are terrestrial and active on the moist soils of the forest, preying or scavenging on ants and other small insects. In this monomorphic species, both sexes have wings and are capable of flight (Figure 1E–H). Females while walking along the tips of grass and low vegetation on the forest floor, signal with rapid single pulse flashes with averaged flash duration 201 ms, flash interval 124 ms. Patrolling males usually fly about 2 m high through the forest and display relatively prolonged, single-pulse, intensity-modulated flashes when searching for females with averaged flash duration 924 ms, interval 2757 ms, and rate 0.28 (flashes/s) [15].

Males and females of both species were lab reared at Huazhong Agricultural University in Wuhan City, Hubei Province, for two generations. The original population of *E. pseudosauteri* was collected by Fu from Mt. Tian Tai, Qionglai County, Chengdu city, Sichuan Province in May 2012, while *Abs. chinensis* was collected in Huangpi County, Wuhan City in July of 2012. For both species larval fireflies were bred in transparent plastic boxes (20 cm diameter, 6 cm high). *E. pseudosauteri* larvae were provided with crushed land snails (*Bradybaena ravida sieboldiana*) as prey [14] and *Abs. chinensis* were provided freshly killed ants *Polyrhachis vicina* Roger, and mealworms *Tenebrio molitor* [15]. Pupae were separated by sex and housed with moist filter paper until emergence.

### 2.2. Sexual Dimorphism, Mating Systems and Reproductive Systems

Elytra and hind wings of *E. pseudosauteri* were examined and photographed by a Nikon D4 digital camera or Nikon SZX 16 stereomicroscope coupled with a DP72 CCD camera. Right elytra of female fireflies of *Abs. chinensis* were cut off to determine degree of dimorphism and photographed by a Nikon D4 digital camera. Length of elytra and hind wings of both two species were measured by vernier calipers.

To confirm whether females are capable of flight, field observations were conducted by Fu during the courtship activity peak from 2010 to 2014 (April to May for *E. pseudosauteri* in Mt. Tiantai, Chengdu City, Sichuan Province and July to August for *Abs. chinensis* in Mt. Sushansi, Wuhan City, Hubei Province).

To investigate whether the species is monandrous or polyandrous, newly eclosed females were assigned randomly to one of three mating treatments: each female was mated to either a single male or mated sequentially to two or three different males (only a single male was introduced at a time). All mating pairs were constantly monitored, and males were removed once they had mated with the female and were replaced by another virgin male. Virgin males were also provided to females 24 h or 48 h after the first copulation.

In previous studies [5] on reproductive traits and spermatophores in fireflies, males usually were fed a 40% sucrose solution with 1% rhodamine B, a thiol-reactive fluorescent dye that forms covalent bonds to protein. This product is known to stain spermatophores [16,17], to visualize portions of the male reproductive tract responsible for producing spermatophore precursors, and to track the location of male spermatophores within the female reproductive tract [9]. However, we found the 1% rhodamine B solution was toxic to both species of male fireflies, and a lower concentration of 0.5% rhodamine B prevented males mating with female fireflies.

To examine male and female reproductive anatomy, adults of each species were frozen at −20 °C in 75% EtOH and stored until dissection [5]. Reproductive tracts removed from males and females were observed with a Nikon SZX 16 stereomicroscope, fat bodies attached to reproductive structures were carefully removed either by a fine brush or by soaking in cold 5% KOH for 1 min and photographed with a DP72 CCD camera.

### 2.3. Time Course of Spermatophore Transfer

Mating experiments conducted with virgin *E. pseudosauteri* and *Abs. chinensis* to characterize the time course of spermatophore transfer and degradation follow Fu et al. [5]. The presence and position of spermatophores within the female reproductive tract were traced at time points ranging from 0 min, 15 min, 30 min, 1 h, 2 h, 6 h, 12 h, 24 h to 48 h after the beginning of mating. Copulations were terminated by freezing, and pairs were stored in EtOH until dissection. Reproductive tracts removed from males and females were observed with a Nikon SMZ1500 stereomicroscope equipped DP72 CCD camera.

## 3. Results

### 3.1. Sexual Dimorphism, Mating Systems

In the E. pseudosauteri female, the hind wing is 2.04 ± 0.05 mm (Mean ± SE) long while elytron is 6.00 ± 0.06 mm (Mean ± SE). Field observation confirmed the female is flightless. Females of E. pseudosauteri are monandrous and did not accept virgin males 24 h after first copulation. Females started to lay eggs 48 h after first copulation.

In the *Abs. chinensis* female, the hind wing is as long as the elytron: hind wing is 6.00 ± 0.02 mm (mean ± SE) long while elytron is 6.00 ± 0.03 mm (mean ± SE). Field observation confirmed the female is capable of flight, and males often chased flying females to mate. Females of *Abs. chinensis* are monandrous and did not accept virgin males 24 h or 48 h after first copulation.

### 3.2. Male Reproductive Systems

In *E. pseudosauteri*, the paired testes are located at the anterior end of the abdomen (Figure 2A,B), each thickly covered by yellow fat body and connective tissue. The tubular seminal ducts lead to the seminal vesicles covered by yellow fat body, which appeared as ovoid enlargements at the proximal ends of these ducts. The seminal vesicles empty into the ejaculatory duct at its junction with the accessory glands. The most conspicuous structures in the reproductive tract of *E. pseudosauteri* males were three pairs of bilaterally symmetrical accessory glands (Figure 2A,B) that enter the ejaculatory duct at a common point. The most central are paired curled glands, which are tapering, folded glands arranged longitudinally in the abdominal cavity and appearing much longer in dorsal view; these glands are approximately 0.1 mm long and measured 0.05 mm at their widest point (Figure 2A, CG). Near the curled glands are two additional pairs of tubular short and wide accessory glands (about 0.15 mm long and 0.05 mm wide) (Figure 2B, SG). The thin-walled long glands are approximately 0.5 mm long and narrow, widening distally to about 0.05 mm and ending in a white, spongy tissue (Figure 2A,B, LG).

In *Abs. chinensis* males, reproductive structures were quite similar to those of *E. pseudosauteri* (Figure 3A,B). Male accessory glands were virtually identical in both species, with only slight variations in the long accessory glands. The tip of the long glands of *E. pseudosauteri* is swollen and round while the tip of the long glands of *Abs. chinensis* is elongate and slender.

### 3.3. Female Reproductive Systems

Interpretations of internal female reproductive anatomy is based on Ballantyne et al. (2011).

*E. pseudosauteri*: the paired ovaries contained ovarioles with oocytes in different developmental stages, with mature oocytes in the lateral oviducts. These lateral oviducts converged into a median oviduct. No sclerotized bursa plates (BP) were observed embedded in the lateral walls of the bursa (BC) (Figure 2C). A sclerotized median oviduct plate (MOP) was observed (Figure 2C,D; Figure 4A). Located at the anterior end of the BC were two structures: a small, spherical spermatheca usually covered by white fat bodies, and a much larger spermatophore digesting gland (SDG). The spermatheca is connected via a short duct to the dorsal side of the BC (Figure 2C,D). Anterior to the spermatheca was the semispherical, thin-walled SDG, which eventually contained the spermatophore as it was being digested (Figure 2C). Posterior to the entry of the median oviduct a small elongate dual branched gland entered the vagina (Figure 2C,D). At present the function of this gland is unknown. For convenience here is it named as a female accessory gland (FAG).

*Abs. chinensis*: the paired ovaries contained ovarioles with oocytes in different developmental stages, with mature oocytes in the lateral oviducts. These lateral oviducts converged into a median oviduct. A sclerotized median oviduct plate (MOP) was observed (Figure 3C,D). Two symmetrical hook-shaped bursa plates (BP) are embedded the dorsal wall of the BC (Figure 3C,D). Located at the anterior end of the BC are two structures: a small, spherical spermatheca usually covered by white fat bodies, and a much larger spermatophore digesting gland (SDG). The spermatheca is connected via a short duct to the dorsal side of the BC (Figure 3C,D). Anterior to the spermatheca is a semispherical, thin-walled SDG, which eventually contained the spermatophore as it was being digested (Figure 3C). Posterior to the entry of the median oviduct a small elongate unbranched FAG entered the vagina (Figure 3C,D).

### 3.4. Time Course of Spermatophore Transfer and Degradation

The *E. pseudosauteri* male spermatophore consists of an outer membranous sheath surrounding a spongy matrix. Within this matrix is a sperm-containing ampulla, which empties into the female spermatheca through a sharp tip through the matrix and outer sheath. Spermatophore transfer was complete within 15 min following copulation initiation (Figure 4A). The spermatophore entered female bursa and the tip of the spermatophore pointed towards the spermatheca. The spermatophore position did not change except that within an hour the tip of the spermatophore was closer to female spermatheca (Figure 4B,C and Figure 5A,B). At 1 h, the pointed tip was within the spermathecal duct (Figure 5B), and the spermatophores remained intact within the female reproductive tract for up to 12 h after mating (Figure 4 and Figure 5A,B). At 24 h, the spermatophore entered the SDG, and by 48 h the male spermatophore had degraded such that only small fragments remained (Figure 5C–F).

The *Abs. chinensis* spermatophore consists of an outer membranous sheath surrounding a spongy matrix. Within this matrix is a sperm-containing ampulla, which empties into the female spermatheca through a sclerotized tubular duct from the top that passes through the matrix and outer sheath (Figure 6B,D). At 15 min after copulation, the spermatophore had not reached the female bursa. Spermatophore transfer was complete at 30 min after copulation. At 30 min, the spermatophore was not within the SDG but a sperm duct had developed from the top of the spermatophore and was inclined towards the spermatheca (Figure 6A,B). The position of the spermatophore did not change and it remained intact up to 24 h (Figure 6C,D). At 48 h, the spermatophore entered the SDG and was digested, and eggs entered the median oviduct (Figure 6E,F).

### 3.5. Overview of Female Anatomy in the Luciolinae

Below we follow generic and specific categories as outlined in Ballantyne et al. [12].

Descriptions of females initially concentrated on external morphology including colour [18,19,20,21,22,23,24,25,26,27,28]. Association of females was based often on label data, similarity of location and colour pattern to that of the male, and occasionally pairs taken in copulo.

As fresh material became available and interest in firefly anatomy increased, the internal reproductive system was investigated [5,9,10,14,15,26,29,30,31,32,33,34,35,36,37,38].

Finally, attempts have recently been made to associate males and females using molecular technology [39].

As a result we now have some information about females from 26 of the 28 genera in the Luciolinae as defined in Ballantyne et al. [12], including often estimations of flight capacity. Of these, nine are characterised from external morphology only [indicates number of species] and include: *Aquilonia* (2), *Atyphella* (11), *Convexa* (1), *Lloydiella* (1), *Inflata* (1), *Magnalata* (1), *Missimia* (1), *Pygatyphella* (10) and *Pacifica* (3). The following genera are characterized from both external morphology and features of the internal female reproductive system [indicates number of species with internal anatomy information]: *Abscondita* (4), *Aquatica* (4), *Asymmetricata* (2), *Australoluciola* (4), *Colophotia* (2), *Curtos* (1), *Emarginata* (1), *Emeia* (1), *Kuantana* (1), *Luciola* s. str. (3), *Medeopteryx* (8), *Pteroptyx* (9), *Pygoluciola* (6), *Pyrophanes* (2), *Sclerotia* (4), *Triangulara* (1), *Trisinuata* (1).

Females from these aforementioned 17 genera are characterized by possession or absence of at least one of the following: median oviduct plate, bursa plates, spermatophore digesting gland and in some cases observation of intact or partially digested spermatophore. Six of these genera have no bursa plates.

## 4. Discussion

McDermott (1966) [40] listed, worldwide, 329 species of Luciolinae from seven genera. Ballantyne et al. [12], working in a more restricted area that excluded Europe, Asia minor and Africa, listed 27 genera and 222 species. Spermatophore production (either as prespermatophores or spermatophores) has only actually been demonstrated, and described, for less than twenty of these species. All other interpretations are inference.

We can determine possible spermatophore production from three aspects:

1.The male has a certain type of accessory gland which produces the appropriate components of a prespermatophore. Hayahsi and Suzuki [8] investigated eight Luciolinae (including *Pristolycus*) and demonstrated prespermatophores in all. They only examined the female reproductive anatomy of one species, viz. *Luciola cruciata* where they saw sperm in the spermatheca “but no obvious spermatophore fragments in their storage organs”. Identifying prespermatophores is a reliable and repeatable way of determining production even if the female is not available.2.Direct observation of the spermatophore in the female reproductive tract. South et al. [9] demonstrated spermatophore production in the female tract of three species and Fu et al. [5] for one.3.Structures in the female reproductive system like bursa plates and a spermatophore digesting gland (assuming it is visible) suggest the receipt of a spermatophore. (We discount the presence of the median oviduct plate as we are unsure of its function). However, this is not always the case as *Luciola cruciata* for example has no bursa plates yet females still receive spermatophores. This involves inference, as we do not have enough information about the occurrence of spermatophores to make any correlations with female reproductive anatomy except for *Pteroptyx maipo*, where the bursa plates hold the spermatophore partly projecting into the digesting gland [33]. Thus, for the 17 genera listed above for which we have some information about their internal female reproductive structures, six genera (*Asymmetricata, Curtos, Kuantana*, some *Luciola* s. str. *Sclerotia* and *Triangulara*) have no bursa plates and, if the digesting gland is not expanded, no inference about spermatophore receipt can thus far be made.

Sexual dimorphism reaches its most obvious expression in many Lampyrinae where the female is flightless and much larger than the male [10,41]. South et al. [10] indicate they support the idea of the development of neoteny (as wing loss) first, followed by the development of flightlessness in the female. We did not find any evidence here that conflicts with this statement. However, of the eight Luciolinae species South et al. [10] listed with confirmed spermatophore production, three (*Hotaria parvula. Luciola kuroiwae* and *Luciola yayeyamana*) were incorrectly listed as flighted when they are flightless. While some Luciolinae flightless females have similar ‘facies’ to the larger Lampyrinae females, none approaches them in size (i.e., length) [30,31,32]. However, thus far it has been difficult to obtain an accurate determination of this size difference for more than a few species. Measurements of length are highly inaccurate since the female abdomen can be greatly distended, and it is possible that the only objective way to compare sizes will rely on mass. South et al. [9] showed a size dimorphism range, measured as average female/male wet mass, of 1.4 to 2.6 for both *Luciola lateralis* and *L. cruciata*. However both of these species have fully winged and flighted females. Here without accurate measurements of mass, and subjective assessments only of ‘similar’ size of males and females in *Emeia pseudosauteri*, we cannot conjecture further on what would constitute a significant size difference.

Here we address females as either flighted or not. There are a number of Luciolinae females where the hind wings are slightly shortened, but without any field observations there is no way of determining if the female can actually fly. The significance of a possible shorter range of flight is not immediately apparent to this argument, apart from restricting species dispersal. Ballantyne and Lambkin [31] addressed female aptery indicating that flight would not be possible with a hind wing length below some unknown threshold value, and Ballantyne and Lambkin [32] indicated the variability existing in the Luciolinae flightless females with respect to fore and hind wing length. Fu’s field observations on *Emeia pseudosauteri* at least have indicated that a hind wing/fore wing length of 1/3 apparently prevents flight in that species.

Multiple male accessory glands are found in many insects, and the roles played by different glands in spermatophore formation have been reported (reviewed by [42,43]). In the Luciolinae, males have three pairs of accessory glands, of which the most prominent are the tightly coiled spiral accessory glands which manufacture the main structural spermatophore component [9]. *P. collustrans* lacks accessory glands and does not form spermatophores [9,17,44] suggested that the presence of prespermatophores in the glands is a reliable indicator of spermatophore production in fireflies. Hayashi and Suzuki [8] examined the male internal reproductive system in 20 Japanese species belonging to 10 genera for the presence or absence of prespermatophores, and suggested males with only one pair of glands did not produce spermatophores. South et al. ([10]) further expanded this with an overview of 32 Lampyridae with *Pyropyga nigricans* having the maximum number, five pairs of glands. They suggested that the loss of glands, or diminution in size of glands, is an indication of the loss of the ability to produce spermatophores. However, it is not possible to determine the minimum number of accessory glands needed to produce viable spermatophores, short of appreciating the essential components produced in the spiral glands. There was no significant difference in size (as a subjective measurement) between the male accessory glands of *Abs. chinensis* (with flighted female) and those of *Emeia pseudosauteri* (with flightless female). There was, however, a slight difference in the shape of the apex of the long glands (described above). Both had three pairs of glands.

The female reproductive systems of the two species examined here are consistent with the few observations on the Luciolinae which have already been made, e.g., [9,10,12,31,32,33].

Spermatophore disintegration is largely complete by 24 h (*E. pseudosauteri*) or 48 h (*Abs. chinensis*) postmating, with only small fragments remaining in the spermatophore-digesting organ. The persistence of the spermatophores within the female reproductive tract for 6 h (*E. pseudosauteri*) and 24 h (*Abs. chinensis*) is not different to that noted by Fu et al. [5] for *Aq. ficta* and they ruled out its persistence as a possible explanation for monandry.

No further investigation of the function of the presumed female accessory gland has yet been undertaken but given its position posterior to the opening of the median oviduct where fertilized eggs are presumably delivered into the vagina, it may function to secrete some sort of egg capsule.

The development of the short duct derived from the spermatophore itself that joins the spermatophore to the spermathecal duct also needs further investigation. It is unclear if the spermatophore is produced with this duct already developed in those species where the duct arises from the side of the spermatophore (unlikely), or that it develops after extrusion in response to some stimulus from its position in the bursa near the spermathecal duct. The orientation of the spermatophore in the bursa would then be of no significance. In *E. pseudosauteri,* the spermatophore is elongate and apically pointed and it is this pointed end which engages with the base of the spermathecal duct. It is not yet clear whether it acquires a pointed end after extrusion and once in the bursa.

## Figures and Tables

**Figure 1 insects-12-00365-f001:**
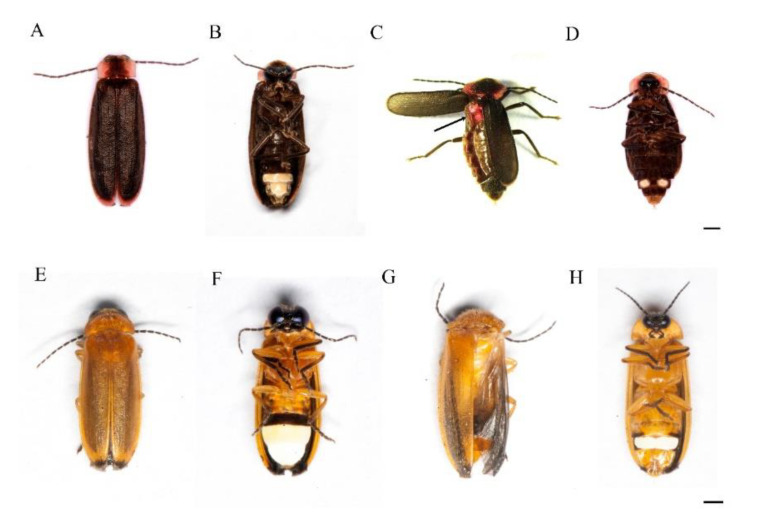
(**A**–**D**) *Emeia pseudosauteri* ((**A**,**B**) adult male; (**C**,**D**) adult female); (**E**–**H**) *Abscondita chinensis* ((**E**,**F**) adult male, (**G**,**H**) adult female). (**A**,**E**) Dorsal. (**B**,**D**,**F**,**H**) Ventral. (**C**) Dorsal showing developed left elytron and very short left hind wing. (**G**) Dorsal showing developed right hind wing with right elytron removed. Scale bar = 1 mm.

**Figure 2 insects-12-00365-f002:**
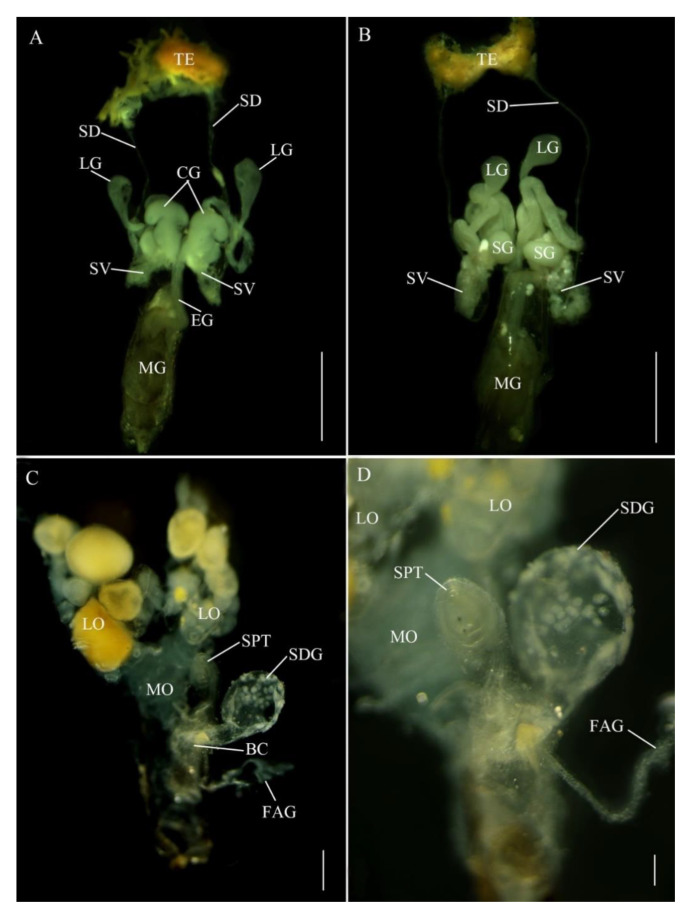
Reproductive system anatomy in *Emeia pseudosauteri* ((**A**,**B**) male; (**C**,**D**) female); (**A**,**C**,**D**) Dorsal. (**B**) Ventral (**D**) Detail in area of spermatheca and median oviduct plate. Figure legend: BC, bursa copulatrix; CG, curled glands; EJ, ejaculatory duct; FAG, female accessory gland; LG, long accessory glands; LO, lateral oviducts; MG, male genitalia; MO, median oviduct; MOP, median oviduct plate; OV, ovaries; SD, seminal ducts; SDG, spermatophore-digesting gland SG, short accessory glands; SPT, spermatheca; SV, seminal vesicle; TE, testes; V, valvifer. Scale bars for (**A**,**B**): 0.2 mm, (**C**): 0.5 mm, (**D**): 0.2 mm.

**Figure 3 insects-12-00365-f003:**
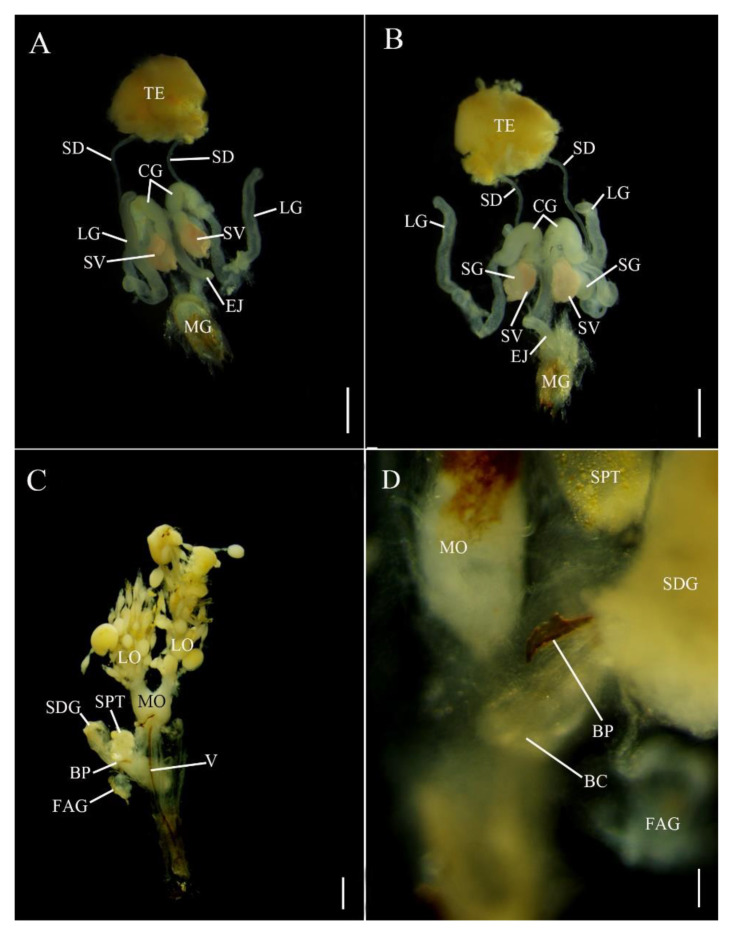
Reproductive system anatomy in *Abscondita chinensis* ((**A**,**B**) male; (**C**,**D**) female). (**A**,**C**) Dorsal. (**B**) Ventral. (**D**) Detail in area of junction of median oviduct and bursa. Figure legends: BC, bursa copulatrix; CG, curled glands; EJ, ejaculatory duct; FAG, female accessory gland; LG, long accessory glands; LO, lateral oviducts; MG, male genitalia; MO, median oviduct; MOP, median oviduct plate; OV, ovaries; SD, seminal ducts; SDG, spermatophore-digesting gland; SG, short accessory glands; SPT, spermatheca; SV, seminal vesicle; TE, testes; V, valvifer. Scale bars for (**A**–**C**): 0.5 mm, (**D**): 0.1 mm.

**Figure 4 insects-12-00365-f004:**
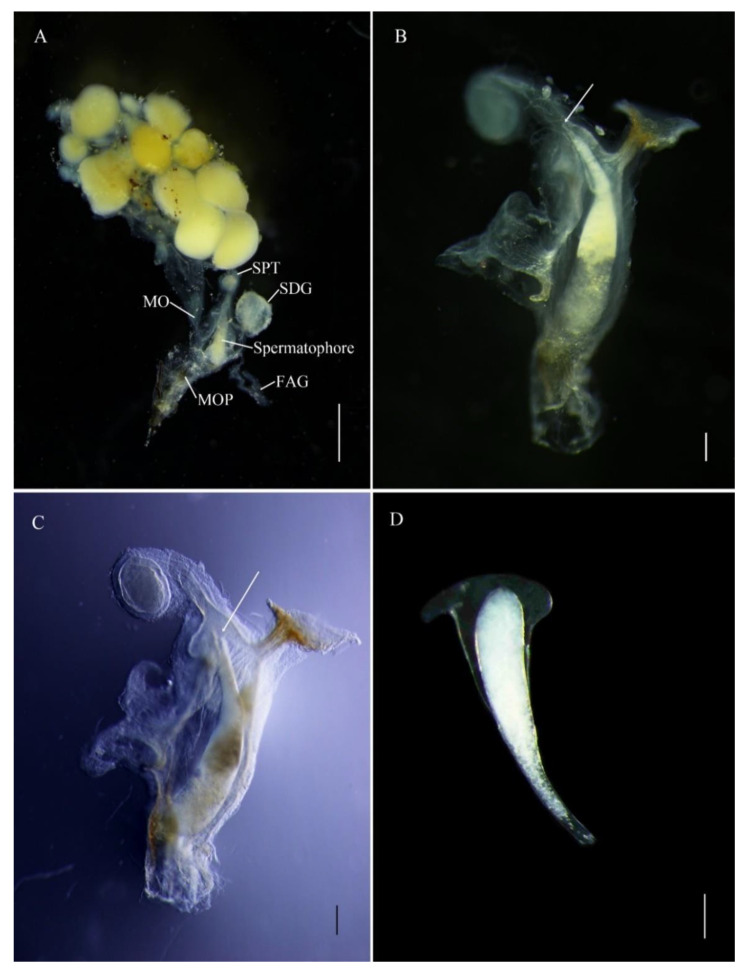
Transfer of male spermatophore to bursa copulatrix of female *Emeia pseudosauteri* at 15 min after copulation. (**A**) Female reproductive system with male spermatophore. (**B**,**C**) White arrows showing tip of spermatophore pointed to spermatheca. (**D**) Intact spermatophore. Figure legend: FAG, female accessory gland; MO, median oviduct; MOP, median oviduct plate; SDG, spermatophore-digesting gland; Spermatophore, male spermatophore; SPT, spermatheca. Scale bars for (**A**–**C**): 0.5 mm, (**D**): 0.2 mm.

**Figure 5 insects-12-00365-f005:**
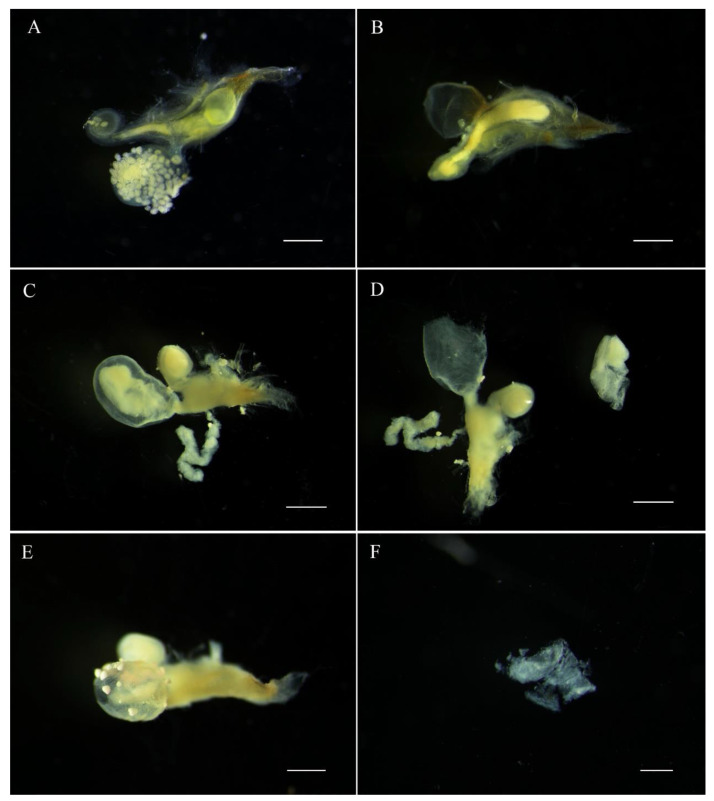
Position and digestion of male spermatophore in bursa copulatrix of female *Emeia pseudosauteri* at various times after copulation. (**A**) Thirty minutes after copulation: tip of spermatophore near spermatheca and not within spermatophore-digesting gland. (**B**) Sixty minutes after copulation: tip of spermatophore has entered spermatheca but not in spermatophore-digesting gland. (**C**,**D**) Twelve hours after copulation: spermatophore partially digested in spermatophore-digesting gland. (**E**,**F**) Forty-eight hours: spermatophore almost digested in spermatophore-digesting gland. Scale bars for (**A**–**E**): 0.5 mm, (**F**): 0.2 mm.

**Figure 6 insects-12-00365-f006:**
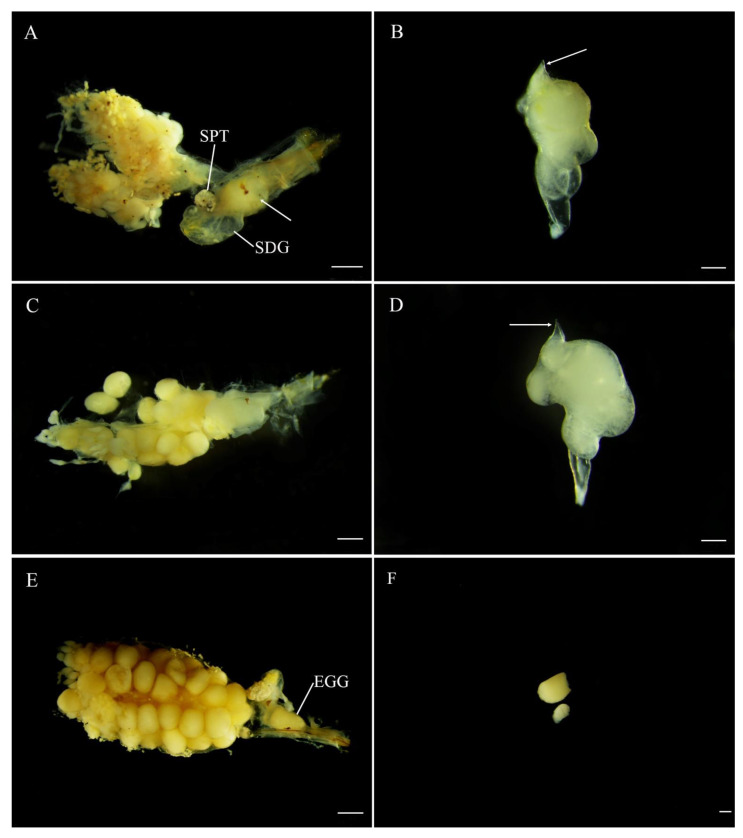
Position and digestion of male spermatophore in bursa copulatrix of *Abscondita chinensis* at various times after copulation. (**A**) Thirty minutes after copulation: female reproductive system with male spermatophore indicated by while arrow in its bursa copulatrix (female accessory gland removed). (**B**) Thirty minutes after copulation: intact male spermatophore dissected from female bursa copulatrix. (**C**) Twenty-four hours after copulation: spermatophore entering spermatophore-digesting gland. (**D**) 24 h after copulation: Intact spermatophore dissected from female bursa copulatrix. (**E**) Forty-eight hours after copulation, spermatophore partially digested in spermatophore-digesting gland. (**F**) Forty-eight hours after copulation: partially digested spermatophore dissected from female bursa copulatrix. Figure legend: SDG, spermatophore-digesting gland; SPT, spermatheca; EGG, mature egg. Scale bars for (**A**,**C**,**E**): 0.5 mm, (**B**,**D**,**F**): 0.2 mm.

## Data Availability

No additional data available.

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
