# Peer review of "Reproductive Systems, Transfer and Digestion of Spermatophores in Two Asian Luciolinae Fireflies (Coleoptera: Lampyridae)"

_insects, 2021, doi:10.3390/insects12040365_

Round 1
Reviewer 1 Report
General comments:
This manuscript reports the internal reproductive anatomy of males and females of two Asian Luciolinae fireflies, Emeia pseudosauteri and Abscondita chinensis. They found that E. Pseudosauteri is sexually dimorphic and A. chinensis is sexually monomorphic. This manuscript provide novel knowledge for the biology of fireflies. I have several specific comments as following:
- In the last sentence of the third paragraph in Introduction, I don’t understand why say “flightless females do not produce spermatophores”? Is spermatophores produced by males?
- Authors determined the length of many parts of wings and reproductive systems. I am not sure whether these values of the length are mean values or just individual. I think mean values should be better and the related variance values. And the same is for time course experiments.
- In the Introduction, authors mentioned the topic about the relationship between spermatophore and sexual seletion. I suggested authors can add more contents about this topic in the Discussion.
Author Response
- English expression. There is some local idiom like the use of a singular noun (firefly is), omission of the indefinite article, and some instances where plural tense is a better choice. Changes are made at:
Lines 112, 149, 152, 154, 180, 181, and at line 384 ‘for’ was inserted.
- your comment 1. Modified to indicate female receive spermatophores.
- Your comment 2 The length of elytra and hind wings of both two species of fireflies were measured by using the method of mean±SE.
- There is just not enough information about mating etc in the Luciolinae(unlike the north American Lampyrinae) to be able to address this yet and that is one of the reasons we have begun to investigate these aspects.

Reviewer 2 Report
This study describes the internal reproductive organs of two firefly species of the subfamily Luciolinae, one with flightless female and the other with flighting female. It is known that the spermatophores transferred from the male to the female are digested and used for nutrients by the female, but such nuptial gifting is lost in the firefly groups with conspicuous sexual dimorphism in body size (much larger female than male, and usually completely wingless female). This study only shows that the spermatophore becomes smaller in the female reproductive organs with a series of photographs of it after copulation.
However, the photographs are not so clear to show spermatophore digestion in the female. The most part of spermatophores seems remain in the bursa for long time. The number of specimens examined (sample size) and any statistical tests are not shown. In the method, the rhodamine B treatment is included, but there are no results and discussion on it. Several references in the text are not included in the reference section and the opposite also occurs. English is not appropriate.
Some examples to be wrong:
3-4: Lampyridae---Lampyridae)
17: examined---is examined
18: Abs.---A.
19-22: Most genera (26/28) of Luciolinae produce spermatophores despite being partial degeneration of female wings.
28: and the many different forms include food items---and include the many different forms such as food items
.
.
80: 1m---1 m
.
.
.
Author Response
We are very concerned with this statement “”Most genera of Luciolinae (26/28) produce spermatophores despite being (sic) partial degeneration of female wings”.
The first part of this statement is false and we hope that this misinterpretation did not skew this reviewer’s ability to fairly assess this paper. The degeneration of wings in the female is not widespread occurring only in three genera.
We are aware of only 9 Luciolinae (included here is Pristolycus) males which produce prespermatophores and four other species for which both male and female reproductive anatomy has been investigated. Most other suggestions about possible spermatophore receipt by the female is made on the observation of bursa plates and spermatophore digesting gland, but not direct observation of an actual spermatophore.
It was on this basis that our investigation was undertaken.
In an attempt to accommodate this apparent misunderstanding/misinterpretation we have added a considerable section of explanation following the new heading for Discussion at lines 292-323.
Your specific comments:
Lines 3-4 Lampyridae). Thank you.
Line 17 a comma after examined, will help indicate the ‘ís’ here is understood.
Line 18 this abbreviation is acceptable and done extensively in this manner in Ballantyne et al 2019 to accommodate genera which begin with the same letter.
Lines 19-20 we disagree. Modified ‘are known to’ to ‘may’ should help here.
Line 28: and the many different forms include food items---and include the many different forms such as food items.
The process of copulation in insects often involves the exchange of a variety of materials from male to female. These may be exchanged prior to, during, or after the process of copulation and the many different forms include food items and spermatophores.
Your modification alters the sense of the sentence. The first sentence refers to a variety of materials males can exchange with the female and the second elaborates on the nature of some of this variety namely food items and spermatophores.
Thank you for your comments on the reference section which has been completely overhauled.

Reviewer 3 Report
The paper is well-prepared and well-presented. It provides some new information about reproductive anatomy of two species that contributes to the interesting evolutionary questions about flightlessness and aptery in fireflies.
My only concern with the ms is the fact that there is no labelled “discussion” section; however, this is possibly due to an omission of a sub-heading.
Some specific comments below
L 56 and 57 I think the sentence was supposed to read “species with flightless females” because females do not produce spermatophores
L 231 the mention of the sclerotised tubular duct refers to Fig 6B & D. Those figures need labelling to show the structure referred to
L 253 Is this where the discussion section starts? If so, it needs a subheading.
L 301 “flighted” rather than “flight”
Author Response
With regard to your specific comments:
L 56 and 57 I think the sentence was supposed to read “species with flightless females” because females do not produce spermatophores. Amended
L 231 the mention of the sclerotised tubular duct refers to Fig 6B & D. Those figures were labelled (arrows added) to show the structure referred to.
L 253 Is this where the discussion section starts? If so, it needs a subheading. Discussion heading inserted at lines 292-323
L 301 “flighted” rather than “flight”. Amended.
Round 2
Reviewer 1 Report
I have no more comments
Reviewer 3 Report
The ms can be accepted for publication